# Similar freezing spectra of particles on plant canopies and in air at a high-altitude site

Annika Einbock[1] and Franz Conen[1]

[1]Department of Environmental Sciences, University of Basel, 4056 Basel, Switzerland

*Correspondence to*: Annika Einbock (annika.einbock@unibas.ch)

**Abstract.** Plant canopies are an important source of biological particles aerosolized into the atmosphere. Certain aerosolized microorganisms are able to freeze slightly supercooled cloud droplets and therefore affect mixed-phase cloud development. Still, spatiotemporal variability of such biological ice nucleating particles (INPs) is currently poorly understood. Here, we study this variability between late summer and leaf shedding on the scale of individual leaves collected about fortnightly from four temperate broadleaf tree species (*Fagus sylvatica*, *Juglans regia*, *Prunus avium* and *Tilia platyphyllos*) on a hillside (Gempen, 650 m a.s.l.) and in a vertical canopy profile of one *Fagus sylvatica* (Hölstein, 550 m a.s.l.) in north-western Switzerland. The cumulative concentration of INPs active at $\geq$ -10 °C ($INP_{-10}$) did not vary significantly between the investigated tree species, but inferred from leaf mass per area and leaf carbon isotopic ratios seemed to be lower on sun as compared with shade leaves. Between August and mid-November, median $INP_{-10}$ concentration increased from 4 INPs $cm^{-2}$ leaf area to 38 $cm^{-2}$ leaf area and was positively correlated with mean relative humidity throughout 24 h prior to sampling (Spearman's r = 0.52, p < 0.0001, n = 64). In 53 of the in total 64 samples collected at Gempen, differential INP spectra between -3 °C and -10 °C exhibited clearly discriminable patterns: in 53% of the spectra, the amount of additionally activated INPs increased persistently with each 1 °C decrease in temperature; the remaining spectra displayed significant peaks in differential INP concentration above -9 °C, most frequently in the temperature interval between -8 °C and -9 °C (21%), and between -7 °C and -8 °C (17%). Interestingly, the three most frequent patterns in differential INP spectra on leaves in Gempen were also prevalent in similar fractions in air samples with clearly discriminable patterns at the high-altitude site Jungfraujoch (3580 m a.s.l., Switzerland) collected during summer in the previous year. These findings corroborate the idea that a large fraction of the airborne biological INP population above the Alps during summer originates from plant surfaces. The inquiry into which parameter, or set of parameters, could affect biological INP populations on both scales – upwind airsheds of high-altitude sites as well as individual leaves – is an intriguing question for further exploration. A first guess is that leaf wetness duration plays a role.

# 1 Introduction

Processes at the Earth's surface and in the atmosphere are interrelated. Ice nucleating particles (INPs) emitted from the Earth's surface, for example, can initiate freezing in clouds at temperatures exceeding ~ -38 °C where in their absence droplets remain liquid (Kanji et al., 2017; Murray et al., 2012). The impact on cloud physics critically depends on both, total INP concentration and population properties such as individual freezing temperatures of the present INPs (Hawker et al., 2021). The dynamics of biological INPs at mixed-phase cloud height are poorly understood (Burrows et al., 2022; Cornwell et al., 2023). This is

partly due to the large variety and heterogeneity of the involved particles as well as spatiotemporal variations in source activities and drivers thereof (Burrows et al., 2022; Cornwell et al., 2023).

Soil organic matter (Conen et al., 2011; Hill et al., 2016; O'Sullivan et al., 2014) and living vegetation (Felgitsch et al., 2018; Hiranuma et al., 2015; Lindemann et al., 1982; Lindow et al., 1978a; Pummer et al., 2012; Seifried et al., 2020) as well as decaying vegetation (Haga et al., 2014; Schnell and Vali, 1976) are major sources of biological INPs. Ice nucleating active

(INA) microorganisms inhabiting these environments are among the most efficient INPs discovered so far and active at temperatures $\geq$ -10 °C (INP$_{-10}$) (Huang et al., 2021). Recently, also INP$_{-10}$ originating from pollen have been identified (Gute and Abbatt, 2020; Kinney et al., 2024; Wieland et al., 2024). On a global scale, considering land cover (Latham et al., 2014) and leaf area (Vorholt, 2012), plant surfaces provide a giant reservoir of INP$_{-10}$ leaking into the atmosphere. Indication for the large contribution of INA microorganisms emitted from plant canopies to the INP$_{-10}$ population at cloud height above western

Europe was revealed through heat treatment of INPs at the High Altitude Research Station Jungfraujoch (JFJ, 46° 32' 53'' N, 07° 59' 02'' E, 3580 m a.s.l.) in the Swiss Alps (Conen et al., 2022). During summer and early autumn, admixture of air from the planetary boundary layer at JFJ is enhanced (Griffiths et al., 2014) and most soils in the surrounding temperate region are covered by plants. Under these conditions, the phyllosphere – plant surfaces in contact with the atmosphere (Vorholt, 2012) - in the airshed upwind presumably contribute the majority of INP$_{-10}$ observed at JFJ.

Ice nucleation active microorganisms associated with the phyllosphere include various gram-negative bacteria (Kim et al., 1987; Lindow et al., 1978b; Maki et al., 1974) and fungal species (Morris et al., 2013; Pouleur et al., 1992). The ability of gram-negative bacteria to nucleate ice is rooted in the expression of IN proteins that can aggregate into assemblies of varying sizes. The largest protein clusters make freezing close to 0 °C possible (Govindarajan and Lindow, 1988; Qiu et al., 2019). Recently, such clustering was also discovered in cell-free INPs shed by the ice-nucleation active (INA) fungus *Fusarium*

*acuminatum* (Schwidetzky et al., 2023) and in ice nucleating macromolecules (INMs) released by pollen of *Betula pendula* (Wieland et al., 2024). Microbial ice-nucleation activity differs between species (Huang et al., 2021) and strains (O'Brien and Lindow, 1988; Yang et al., 2022; Yankofsky et al., 1981) and may allow the most efficient strains to draw water vapour from the atmosphere into zones of a leaf surface that would otherwise remain dry (Einbock and Conen, 2024). In addition, variations in microbial growth and environmental conditions can substantially change freezing characteristics of microbial INA

populations (Hirano and Upper, 1989; Lindow et al., 1982; Nemecek-Marshall et al., 1993; O'Brien and Lindow, 1988; Richard et al., 1996; Ruggles et al., 1993; Yang et al., 2022). These variations in expression of ice-nucleation activity, the

number of known and perhaps unknown INA species, together with uncertainties concerning their cultivability, make it challenging to predict atmospheric INP dynamics from the assessment of microbial community compositions on leaf surfaces. A complementary approach to understand the link between INPs in phyllosphere and atmosphere is to compare differential INP spectra in both spheres. Here, we investigate INP spectra on leaves of four deciduous tree species from late summer throughout leaf senescence. We discuss trends in cumulative $INP_{-10}$ concentrations and patterns in differential INP spectra. Finally, we compare differential INP spectra on foliage with observations of INP spectra during summer 2022 in air at JFJ.

## 2 Methods

### 2.1 Foliage sampling

We sampled leaves of four broadleaf tree species commonly found in temperate forests near Gempen (GEP, 47° 28' 53'' N, 7° 39' 30'' E, ~650 m a.s.l.) and Hölstein (HOL, 47° 26' 17'' N, 7° 46' 37'' E, ~550 m a.s.l.), Switzerland. Both sampling locations are situated on hillsides in the northern Jura mountains at a linear distance of about 10 km from each other. Sampling was conducted between early August and mid-November, 2023. Samples from Gempen were collected approximately fortnightly on eight occasions. In Hölstein, we collected foliage twice, with five weeks in-between (05.09. and 11.10.). Samples were collected between 09.30 am and 11.30 am local time, stored at 5 °C, and analysed for their INP concentration within three to four days.

In Gempen, we sampled leaves of two *Tilia platyphyllos*, *Fagus sylvatica* and *Juglans regia*, respectively, along a 1.5 km path through a small forest surrounded by agricultural land. Additionally, two *Prunus avium* were sampled around 50 m into an adjacent meadow. We sampled trees least overgrown by taller canopies of other species at more open parts of the path and aimed for leaves least likely affected by throughfall from canopies above. From September 26 onwards, roughly the onset of leaf coloration, the same trees were sampled on successive occasions. The first three sampling occasions, various *T. platyphyllos* (TP1, TP2), *F. sylvatica* (FS2) and *J. regia* (JR2) had been sampled in the same segments of the path as from September 26 onwards. Foliage was cut off trees about 1.5 m to 2.5 m above ground using a pair of flame-sterilized scissors and directly transferred into polyethylene zip bags without further contact. Blank measurements had shown the zip bags did not contain $INP_{-10}$. In Hölstein, covering the entire vertical extent of the canopy, foliage of a *F. sylvatica* was collected at about 10 m, 20 m and 30 m above ground. Leaves were accessed from the Swiss Canopy Crane II (Kahmen et al., 2022) and sampled from the outermost canopy parts. At each height, a sample was taken from the south and the north-facing side.

Meteorological data and atmospheric pollen concentrations were obtained from the nearest stations operated by the Swiss Federal Office of Meteorology and Climatology MeteoSwiss (Binningen, 47° 32' N, 7° 35' E, 316 m a.s.l. and Basel, 47° 34' N, 7° 35' E, 256 m a.s.l., respectively) 10 km NE of Gempen and the Swiss Federal Institute for Forest, Snow and Landscape Research (Hölstein, 47° 26' N, 7° 47' E, 530 m a.s.l.). Due to the difference in altitude between Binningen and Gempen, temperature was corrected by 0.65 K/100 m. Precipitation data at Gempen was obtained from the data portal of MeteoSwiss.

### 2.1.1 Leaf colour

Leaf colour was determined right after sampling by visually matching adaxial leaf sides to one of 2050 reference colours (NCS
Index 2050) provided by the Natural Colour System (NCS). The NCS is a three-dimensional colour model based on human
colour perception (Hård and Sivik, 1981) and has been used to assess vegetation colour in the past (Grose, 2014, 2016; Shen
et al., 2022; Xing et al., 2019). Briefly, blue, red, yellow and green are considered elementary hues that are represented on a
colour circle. Two neighboring elementary hues are mixed in varying proportions to describe the hue of a colour of interest.
In addition to a specific hue, each colour is assigned a nuance which depends on its black-/whiteness and chromaticness (Hård
and Sivik, 1981).

Colours were assessed on sampling days around noon beneath an east-facing window. All leaves comprising a sample were
considered collectively and the main leaf colour identified as colour covering the largest leaf area per sample. Colour
determination was performed by one, or two independent observers. For the colour assessment which took a few minutes per
sample, leaves were removed from the polyethylene zip bags and spread adaxial side up on top of a new polyethylene zip bag
cleaned with 2-Propanol. Samples of approximately 100 cm$^2$ leaf area each (81.0 cm$^2$ to 142.7 cm$^2$) corresponding to between
1 and 19 leaves were stored in 50 mL polypropylene tubes (Cellstar®, greiner bio-one, Switzerland) at 5 °C until INP analysis.
Throughout the process, leaves were handled with gloves (Vasco® Nitril light, B. Braun, Switzerland) cleaned with 2-
Propanol. For statistical analysis, NCS hue labels were converted to a length value as described by Xing et al. (2019). For
visualization in Fig. 3, Fig S1 and Fig S2, NCS values were converted to HEX colour code using an online tool
(https://www.w3schools.com/colors/colors_converter.asp).

### 2.1.2 INP analysis

Ice nucleating particle concentrations were quantified in leaf washing water. Immediately before each freezing test, the tubes
containing the leaf samples were filled up with 50 mL of 0.1% NaCl in Milli-Q® water, sonicated for 1 min (RK 100, Bandelin,
Germany) and shaken manually for another 20 s. Sonication for 1 min was a compromise informed by additional trials. When
compared to different sonication times between 10 s and 6 min, 1 min was long enough to suspend the majority of INPs from
the leaf surface while avoiding visible leaf damage that may promote the release of INPs from within the leaf structure. Leaves
were removed and pressed between sheets of thick cellulose paper (Clairefontaine goldline, 300 gm$^{-2}$, the Netherlands) until
further processing. A total of 28.8 mL of leaf washing water was transferred to two subsets of 72 Safe-Lock tubes (Eppendorf,
0.5 mL, Germany), each containing 200 µL of sample. The two subsets were cooled in parallel in two separate cold baths
(Lauda RC6, Lauda-Königshofen, Germany) from -3 °C to -10 °C. After every 1 °C step in cooling (rate 0.3 °C min$^{-1}$),
temperature was left unchanged for at least 30 s before the number of frozen droplets was determined visually by one observer
on both subsets, or two independent observers on one subset each. When a large fraction of droplets had frozen at -10 °C,
additional 1:10 and 1:100 dilutions of the leaf washing water were analysed. Where dilutions overlapped, we selected the data
points from the dilution in which the fraction of frozen droplets was closest to 50%. In cases where the pattern of a differential

INP spectrum would have changed solely because of a switch between dilutions, this switch was implemented at the next possible temperature interval. Blanks consisting of the utilized 0.1% NaCl Milli-Q® water did not freeze within the investigated temperature range. Differential INP concentrations were quantified according to Eq. 8 in Vali (2019) and cumulative INP concentrations were calculated as the sum of differential INP concentrations (Vali, 1971, 2019).

### 2.1.3 Leaf mass per area, carbon to nitrogen ratio and $\delta^{13}C$ values

Samples were dried collectively at 40 °C until their weight had stabilized. Leaf mass per area (LMA) was calculated as leaf dry weight divided by leaf area as determined with the software WinDIAS 3 (Version 3.3.0.39, Delta-T Devices, United Kingdom). Leaf carbon (C) and nitrogen (N) concentration as well as stable carbon isotope ratio ($\delta^{13}C$, relative to Vienna Peedee Belemnite (VPDB)) were analysed on an elemental analyzer coupled with an isotope ratio mass spectrometer (EA-IRMS, Integra2, Sercon, Crewe, United Kingdom). For that purpose, a number of randomly selected punches (2 mm diameter)

were taken from each dried sample and weighed into a tin capsule (0.72 mg to 1.68 mg per sample). Mass calibration for C and N quantification was performed with the lab standard EDTA (41.09% C; 9.59% N). For isotopic (C) size correction and calibration the lab standard EDTA (41.09% C; 9.59% N), USGS61 (Coplen, 2019b) and USGS40 (Coplen, 2019a) were used. Carbon to nitrogen ratios (C:N ratios) are shown as atomic ratios. According to the analysed standard substances, the standard deviation for $\delta^{13}C$ is < 0.3‰.

### 2.2 Aerosol sampling and analysis

In a preceding study at JFJ we had collected on 23 days, between the beginning of July and mid-August 2022, a total of 133 aerosol samples (Einbock, 2023). The observatory JFJ is located at a 3 km higher elevation than the foliage sampling sites and situated about 110 km SSE of Gempen. About one third of Switzerland is covered by temperate forest and another third is agricultural land (Beyeler et al., 2021). Sampling at JFJ was conducted with a high-flow rate impinger (flow rate 300 L min$^{-1}$,

Coriolis u, Bertin Technologies, France). Ambient aerosol particles were collected into 15 mL of ultrapure water (W4502-1L, Sigma-Aldrich) containing 0.5% NaCl to reduce osmotically induced stress in biological cells (Stopelli et al., 2014). The majority of samples (n = 129) were collected throughout 30-min periods consisting of five consecutive 5-min intervals. For the remaining samples (n = 4), the impinger was operated at three consecutive 5-min intervals. Water lost during impinger operation was replenished between the 5-min sampling intervals with NaCl-free ultrapure water. After the last 5-min interval,

liquid volume in the sampling cone was quantified. Samples were analysed with the automatic freezing detection apparatus LINDA (Stopelli et al., 2014) immediately after collection in 52 tubes (0.5 mL Eppendorf Safe-Lock, Germany), each containing 200 µL of impinger liquid. All analysed blanks of NaCl-containing and NaCl-free ultrapure water did not contain INPs active above -12 °C. As for the foliage samples, differential INP concentrations were calculated based on Eq. 8 in Vali (2019) and cumulative INP concentrations displayed as sum of differential INP concentrations (Vali, 1971, 2019). Sodium

analysis of impinger liquid (940 Professional IC Vario, Metrohm, Switzerland) revealed that its concentration decreased to 70% of its initial concentration during the 25 min the impinger was operated. This decrease indicates some loss of liquid in

form of droplets; a loss of liquid in form of water vapour only would not have removed NaCl. Therefore, we corrected INP concentrations for potential INP loss with droplets exiting the impinger during collection by multiplying uncorrected INP concentrations with a factor of $\frac{2}{1+0.7}$, assuming the loss of INPs is zero at the beginning of sampling, when there are no INPs
in the liquid, and increases linearly with time. I.e., the fraction of INPs lost with droplets is half the fraction of NaCl lost.

### 2.3 Statistically significant peaks in differential INP spectra

In immersion freezing experiments, it can be presumed that the number of INPs per droplet activated within each investigated temperature interval is Poisson distributed (Vali, 1971, 2019). Accordingly, the standard deviation for the number of INPs activated within each temperature step equals the square root of the number of freezing events in the assay within this
temperature interval (Vali, 2019). We consider a temperature interval ($\Delta T$) in a differential INP spectrum to constitute a significant peak, if the sum of the standard deviations for the number of freezing events in $\Delta T$ and $\Delta(T+1)$ is smaller than the difference in the number of freezing events between $\Delta T$ and $\Delta(T + 1)$, where $\Delta(T + 1)$ is the next colder temperature interval to $\Delta T$.

### 3 Results and Discussion

**3.1 Cumulative INP concentrations**

### 3.1.1 Temporal trends in INP$_{-10}$ concentrations, meteorological parameters and leaf traits

We found a total of 273'295 INP$_{-10}$ on 8'304 cm$^2$ of leaf area (LA) and 733 INP$_{-10}$ in 968 m$^3$ of air (local conditions). From the beginning of August until mid-November, median INP$_{-10}$ concentration on leaves collected at GEP increased from 4 per cm$^2$ of LA to 38 per cm$^2$ of LA. Mean relative humidity (RH) throughout the 24 h prior to sampling increased from around
60% to 82% with a transient maximum of 73% on September 26. Mean air temperature throughout the 24 h prior to sampling decreased from about 20 °C in August to approximately 10 °C in mid-October (Fig. 1). There was no precipitation 24 h prior to sampling during the entire campaign at both sites, except for the last sampling day in November (17.3 mm over 24 h). Cumulative INP$_{-10}$ concentrations correlated significantly with mean RH (Spearman's r = 0.52, p < 0.0001, n = 64) and mean air temperature (Spearman's r = -0.34, p = 0.006, n = 64), respectively. The correlation between INP$_{-10}$ concentration and mean
RH persisted when considering the tree species separately, except for *F. sylvatica*. On the level of individual tree species, the negative correlation between INP$_{-10}$ concentration and mean air temperature was only significant in *P. avium*.

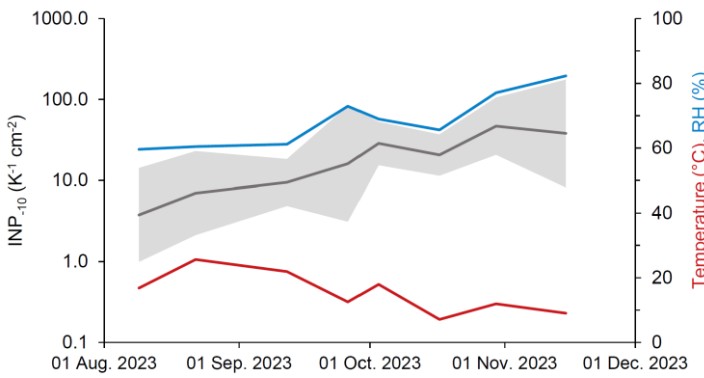

**Figure 1:** Temporal development of the median INP$_{-10}$ concentration on leaves collected in Gempen (grey line), relative humidity (blue) and air temperature (red). The multiplicative standard deviation (Limpert et al., 2001) of the INP$_{-10}$ concentration is indicated as shaded area.

The activation temperature ($\geq$ -10 °C) of the investigated INPs suggests they are predominantly of biological origin (Huang et al., 2021; Kanji et al., 2017). The analysed sample type (leaf washing water of broadleaf trees) further renders a considerable contribution of INA microorganisms to the investigated INP population likely (Hill et al., 2014; Lindow et al., 1978a). Potential additional sources of INP$_{-10}$ at the sampling locations known at present might include INMs derived from pollen (Gute and Abbatt, 2020; Kinney et al., 2024; Wieland et al., 2024). Tree species commonly found in central European forests, including the four species investigated here, typically do not flower between August and November (Anonymos, 2002). Still, wind-dispersed pollen from other plant species and long-range transport deposited onto the leaves might have contributed to the analysed INP population. Atmospheric pollen concentrations at Basel are continuously monitored and measurements for about 50 plant species or families are available until 29.09.2023, seven among them until the end of the sampling period. While cumulative INP$_{-10}$ concentrations on leaves increased, airborne pollen concentrations showed a decreasing trend. There was no correlation between pollen concentrations from individual species or families and INP$_{-10}$ concentrations. In the following, we will therefore focus on microbial INPs.

Generally, the cumulative INP$_{-10}$ concentration in the bacterial population on foliage is affected by both, population size and the frequency of INPs among cells (nucleation frequency NF) (Lindow et al., 1982). Temperature (Hirano and Upper, 1989; Nemecek-Marshall et al., 1993; Ruggles et al., 1993), RH (Hirano and Upper, 1989; Leben, 1988), nutrient availability (Nemecek-Marshall et al., 1993; Ruggles et al., 1993) and plant genotype (Lindow et al., 1978a; O'Brien and Lindow, 1988), among other factors, have been found to influence population size and NF of epiphytic bacteria, sometimes in an opposing manner. Culture age and variations in growing conditions were further found to influence the ice nucleation activity of different *Fusarium* species (Richard et al., 1996; Yang et al., 2022).

In this study, decreasing air temperatures between late summer and autumn might have triggered an enhanced expression of IN proteins in microbes (Anderson et al., 1982; Hirano and Upper, 1989; Nemecek-Marshall et al., 1993), possibly contributing to the observed increase in INP$_{-10}$ concentrations. The significant correlation between air temperature and INP$_{-10}$ concentration in *P. avium* and the absence of such a correlation in the other species might have been related to the free-standing position of

*P. avium* in a meadow while the other trees investigated were situated within a forest. Thus, the *P. avium* were less shielded by surrounding trees against radiative cooling during clear nights and might have experienced lower leaf temperatures at night, pronouncing the effect of temperature on the observed $INP_{-10}$ concentrations. Jordan and Smith (1994) found leaf temperatures during clear nights were around 4 °C below air temperature.

Moisture promotes the survival and growth of epiphytic microorganisms (Beattie and Lindow, 1995; Grinberg et al., 2019), affects their spatial distribution within the leaf microhabitat (Doan et al., 2020) and possibly influences the composition of the phyllosphere microbiome (Beattie, 2011). Intense rain events were found to trigger an increase in the population size of the bacterial INA strain *Pseudomonas syringae* pv. syringae on snap bean leaflets (Hirano et al., 1996). Further, high RH fosters the abundance of *Pseudomonas syringae* in the phyllosphere (Leben, 1988) and seems to enhance ice nucleation activity on foliage (Hirano and Upper, 1989). The strong correlation between RH and $INP_{-10}$ concentration observed here indicates an effect of RH on either the abundance of INA microorganisms on foliage, their NF or both, even though relationships cannot fully be disentangled due to the co-occurrence of continuous trends in $INP_{-10}$ concentrations and meteorological parameters. Increasing INP supply in the phyllosphere at elevated RH might, besides differences in emission mechanisms, contribute to the enhanced concentration of airborne INPs observed under high RH conditions (Testa et al., 2021; Wright et al., 2014).

On the seasonal scale, changes in climate but also leaf characteristics and plant defence during senescence have been associated with shifts in phyllosphere microbial communities in various ecosystems (Kinkel, 1997; Šigutová et al., 2023; Stone and Jackson, 2021). During senescence, leaf processes change and the nutrient content in foliage decreases considerably (Lim et al., 2007). Here, leaf coloration was mirrored in a distinct shift to larger C:N ratios and a smaller amount of N per leaf dry weight in all sampled trees (Fig. S1). Significant correlations between $INP_{-10}$ concentration and leaf colour NCS code expressed as length value (section 2.1.1) (Spearman's $r = 0.29$, $p = 0.02$, $n = 64$), C:N ratio (Spearman's $r = 0.39$, $p = 0.002$, $n = 64$) and amount of N per leaf dry weight (Spearman's $r = -0.41$, $p = 0.0008$, $n = 64$) were found for the entire GEP data set, but were absent when green and coloured leaves were assessed separately ($p > 0.05$, $n = 38$ and $n = 26$, respectively).

### 3.1.2 Differences within and between trees

Cumulative $INP_{-10}$ concentrations did not vary significantly between the investigated tree species. Rather, concentrations differed between some individual trees, e.g. between the two *F. sylvatica* and the two *T. platyphyllos* sampled in GEP (Fig S2).

In the vertically sampled *F. sylvatica* (HOL), $INP_{-10}$ concentrations increased from the top to the lowest part of the canopy. The four samples from the top had a median value of 4.0 $INP_{-10}$ cm$^{-2}$ with a multiplicative standard deviation of $^{x}/$ 1.5 (i.e., 68% of leaf samples at the top of the tree had an $INP_{-10}$ concentration between 2.7 (4.0 / 1.5) and 6.0 (4.0 x 1.5) $INP_{-10}$ cm$^{-2}$). On the first sampling day, $INP_{-10}$ concentrations were highest in the sample from the lowest position within the canopy and facing a SE direction, with concentrations 10 times (73.1 $INP_{-10}$ cm$^{-2}$) as high as at the opposite side of the canopy on the same height. Five weeks later, INP concentrations at the lowest position facing a SE direction had decreased distinctly from 73.1 $INP_{-10}$ cm$^{-2}$ to 15.8 $INP_{-10}$ cm$^{-2}$ but were still twice as high as at the opposite canopy side. These differences might be attributed

to variations in leaf microclimate or localized plant reactions. As expected from previous findings (Bachofen et al., 2020; Matyssek et al., 2010), LMA was greater at the tree top (94.7 g m$^{-2}$ ± 6.8 g m$^{-2}$) compared to the lowest part of the canopy (70.1 g m$^{-2}$ ± 15.2 g m$^{-2}$, n = 4). Likewise, $\delta^{13}C$ values increased with height in the canopy (bottom -29.3‰ ± 0.8‰, top -27.8‰ ± 0.4‰, n = 4, respectively), similar to what has been reported for temperate forests before (Garten and Taylor, 1992; Hanba et al., 1997; Schleser, 1990).

Leaf mass per area and $\delta^{13}C$ covaried in both, the HOL and GEP data set. The concentration of INP$_{-10}$ tended to be higher on leaves with lower LMA and lower $\delta^{13}C$ values (Fig 2). Both parameters did not correlate with leaf N content. Differences in LMA within broadleaf trees result primarily from variations in light availability (Matyssek et al., 2010). In the absence of variations in the atmospheric $\delta^{13}C$, leaf $\delta^{13}C$ in C$_3$ plants is largely determined by stomatal aperture and photosynthetic rate (Farquhar et al., 1982). Enhanced $\delta^{13}C$ values are observed when photosynthetic rates are high or stomates closed, e.g. under elevated light intensity or water stress (Farquhar et al., 1982; Schleser, 1990; Waring and Silvester, 1994).

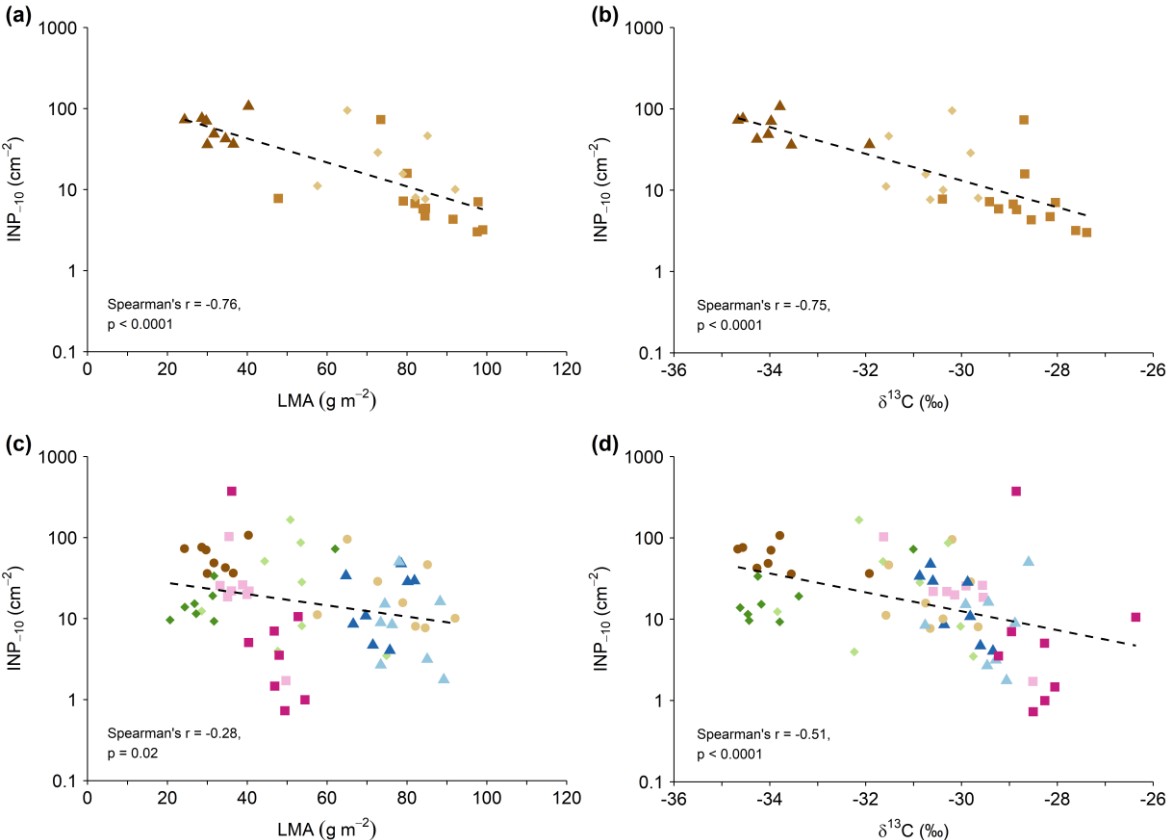

**Figure 2:** Cumulative INP$_{-10}$ concentration versus leaf mass per area (LMA) (a, c) and leaf $\delta^{13}C$ values (b, d) for the GEP data set (c, d) and all *F. sylvatica* samples (a, b). Colours and symbols represent tree species (brown circles: *F. sylvatica*, blue triangles: *P. avium*, green diamonds: *J. regia*, red squares: *T. platyphyllos*), light colours for first and dark colours for second tree sampled (c, d). *F. sylvatica* sampled at Hölstein is represented by squares in the top panels (a, b).

The change in INP$_{-10}$ concentrations with LMA and $\delta^{13}$C might be linked to morphological and physiological differences between sun and shade leaves or differences in microhabitat and -climate that are related to variations in LMA and $\delta^{13}$C. Our

results from Hölstein indicate that INP$_{-10}$ concentrations are lower on leaves more exposed to sunlight in a canopy. Such leaves at the upper and outer part of a canopy might be in general more exposed to stressors or processes potentially removing INPs from leaves compared to leaves in other parts of the canopy. Additionally, gradients in parameters such as RH within forest canopies (Zahnd et al., 2023) might lead to decreasing INP$_{-10}$ concentrations towards the canopy top.

**3.2 Differential INP concentrations**

**3.2.1 Spectral types on foliage**

Of the 64 foliage samples collected in GEP, 53 displayed two clearly discriminable patterns in differential INP spectra between -3 °C and -10 °C. In 28 samples, differential INP concentrations increased persistently with each 1 °C step in cooling (monotonous spectral type). The remaining 25 samples exhibited one (n = 23) or two (n = 2) significant peaks (section 2.3) at

270 temperatures ranging from -3.5 °C to -8.5 °C (numbers indicate the centre of 1 °C temperature intervals implemented in the assays). Most of these peaks occurred around -8.5 °C (11 peaks) and -7.5 °C (9 peaks). Significant peaks at warmer temperatures were more rare (3 peaks each at -6.5 °C and -4.5 °C, 1 peak at -3.5 °C). Generally, discontinuous differential INP spectra indicate that a sample contains distinct INP populations (Vali, 1971). Concurrently, the fraction of INPs among aerosol particles is higher at greater supercooling and the number of airborne INPs was often found to increase roughly exponentially

with decreasing temperatures (Fletcher, N. H., 1962; Kanji et al., 2017; Li et al., 2022). This often-described relationship is equivalent to the monotonous spectral type displaying a persistent increase in differential INP concentrations with progressing cooling.

According to their efficiency, bacterial INPs are categorised as either type I, II or III, corresponding to a decrease in nucleation temperature. Type I INPs are active above -5 °C while type III INPs nucleate ice only below -7 °C. Type II INPs are typically

less abundant and induce freezing between -5 °C and -7 °C (Yankofsky et al., 1981). Five of the seven spectra exhibiting peaks at temperatures > -7.5 °C occurred in *J. regia* (Fig. 3). In contrast, such type I and II modes were never found in *F. sylvatica* and only once in *T. platyphyllos* and *P. avium*, respectively. Samples from *F. sylvatica* predominantly featured the monotonous spectral type. This applied to *F. sylvatica* in GEP as well as in HOL. Peaks in differential INP spectra in *P. avium* tended to be at slightly colder temperature and the monotonous spectral type more abundant than in *T. platyphyllos*. These differences

indicate that variations in leaf habitat properties such as microclimate, leaf morphology and physiology, and cooccurring differences in the phyllosphere microbiome between tree species might have contributed variation to the distribution of spectral types among species. Overall, however, the pattern of differential freezing spectra and INP$_{-10}$ concentration differed widely between samples even within the same species. When combined, spectra of all species sampled on a particular day did also show the two discriminable patterns in differential INP spectra (Fig. S3).

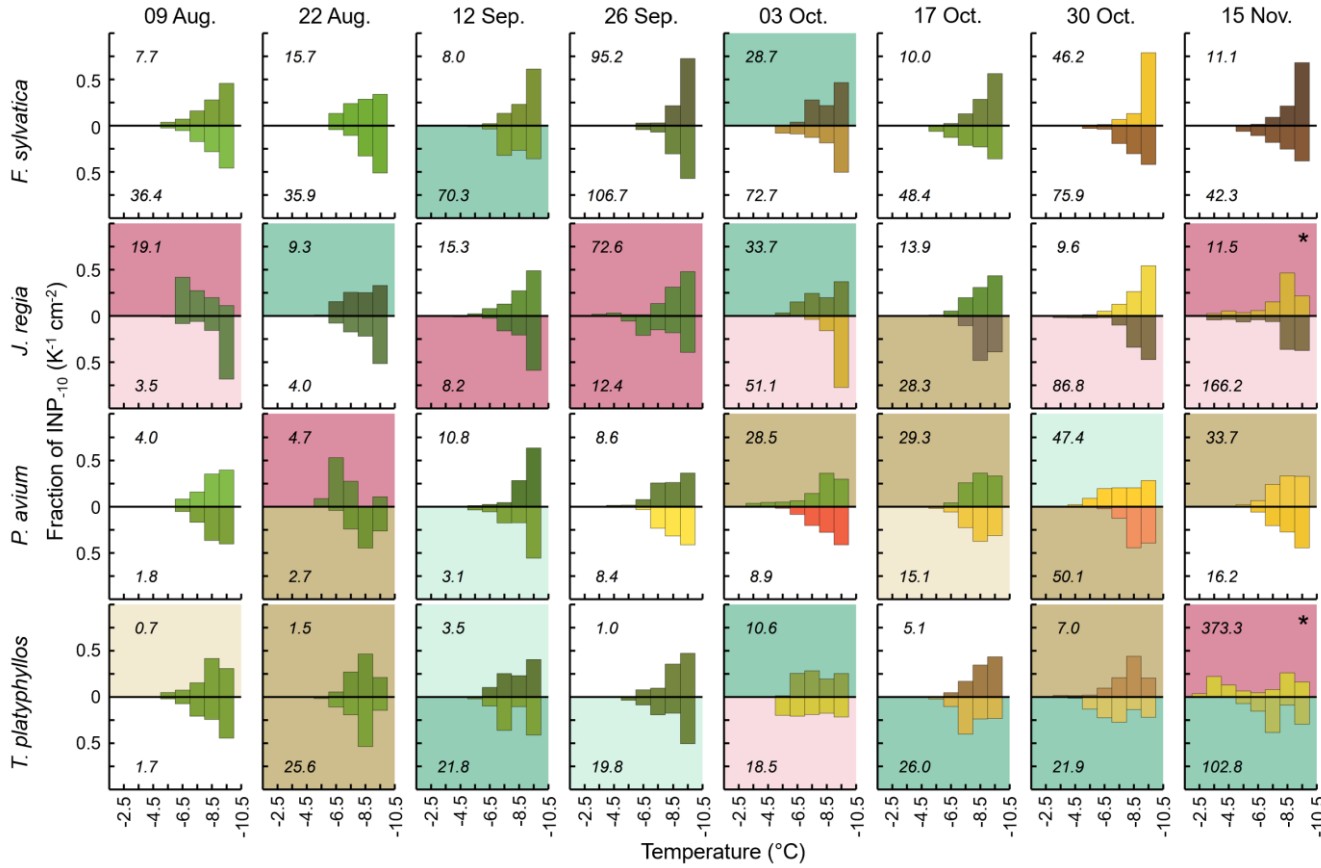

**Figure 3:** Differential INP concentrations (1 K temperature intervals) for tree species (rows) sampled in Gempen on all sampling days (columns), normalized to cumulative INP$_{-10}$ concentration (numbers in the panels). Both samples per tree species and date are shown (above and below x-axis). Colours reflect leaf colours, panel background colour indicates the spectral type (white: monotonous spectral type, brown: peak at -8.5 °C, green: peak at -7.5 °C, purple: peak > -7.5 °C). Dark brown, dark green and dark purple panel backgrounds are for spectra with significant peaks, light brown, light green and light purple panel backgrounds are for spectra with insignificant peaks (section 2.3). The asterisk marks spectra with two significant peaks: one peak >-7.5 °C and one peak at -8.5 °C.

Cumulative INP$_{-10}$ concentrations did not vary systematically between spectral types (Kruskal-Wallis Test, $p > 0.05$). Significant peaks at temperatures > -7.5 °C were observed at total INP$_{-10}$ concentrations as low as 4.7 cm$^{-2}$ leaf area and significant type I peaks at total INP$_{-10}$ concentrations as low as 8.2 cm$^{-2}$ leaf area. This indicates that the occurrence of highly efficient INPs was not solely dependent on the total amount of INP$_{-10}$.

Studies suggest that the expression and arrangement of IN proteins is substantially influenced by environmental conditions and microhabitat characteristics, including differences between host plant species (Hirano and Upper, 1989; Lindow et al., 1982; O'Brien and Lindow, 1988; Ruggles et al., 1993; Yang et al., 2022). For example, UV and γ-radiation (de Araujo et al., 2019; Govindarajan and Lindow, 1988), desiccation (de Araujo et al., 2019), decreasing pH (Lukas et al., 2022) and high temperatures around 30 °C (Nemecek-Marshall et al., 1993) seem to trigger the dissolution of IN protein complexes, whereas

lower temperatures around 15 °C and nutrient limitation can promote their formation (Nemecek-Marshall et al., 1993; Ruggles et al., 1993).

The larger an IN protein cluster and the higher the resultant nucleation temperature of the proteinaceous INP, the faster its activation temperature is lowered by certain stressors, e.g. radiation (Govindarajan and Lindow, 1988). Once exposed to such a stressor, an initially efficient INP will nucleate ice only at colder temperatures (Ruggles et al., 1993). The gradual breakup of larger IN protein clusters under continued stress in unfavorable conditions, or the inhibition of INP cluster formation could be one possible explanation for the development of the monotonous spectral type. This explanation would imply that conditions for the expression and aggregation of IN proteins were less suitable on *F. sylvatica* as compared to the other investigated tree species. Another explanation could be host-specific properties of *F. sylvatica* resulting in a limited set of INP-producing microorganisms that generate the monotonous spectral type only.

Overall, none of the investigated leaf traits seems to explain variations in spectral patterns between leaves or tree species. Yet, other differences in microhabitat, for example due to variations in cuticle characteristics, leaf wettability, leaf exudates or leaf topography (Yan et al., 2022), might have contributed to the unequal distribution of spectral patterns between tree species. These factors can impact the availability of water or nutrients on leaf surfaces, thereby affecting population size and perhaps also composition of epiphytic (INA) microorganisms. In addition, there is a link between nutrient availability and the frequency at which ice-nucleating activity is expressed in certain INA-species (Nemecek-Marshall et al., 1993; Ruggles et al., 1993). As described above, environmental parameters such as temperature and moisture affect INA microorganisms (Hirano and Upper, 1989; Leben, 1988). These parameters can vary on short spatial scales within single canopies (Batzer et al., 2008; Körner and Hiltbrunner, 2018) and variations in microclimate as well as localized plant reactions might have contributed to variations in spectral patterns between leaves of the same species. In summary, the abundance of $INP_{-10}$ and the pattern of differential freezing spectra seem to be less influenced by intrinsic leaf properties than by external conditions.

### 3.2.2 Comparison between foliage and air samples in terms of INPs

The median concentration of 16 $INP_{-10}$ observed per $cm^2$ of leaf surface in GEP was equivalent to the median $INP_{-10}$ concentration in 3.2 $m^3$ of air at Jungfraujoch. Interestingly, the three most frequent spectral types among the clearly discriminable patterns in GEP - the monotonous spectral type, and the spectral types with significant peaks at -8.5 °C and -7.5 °C - were also prevalent in similar proportions in air samples with clearly defined spectral patterns at Jungfraujoch (Fig. 4). This consistency lends support to the hypothesis that plant surfaces contribute the majority of $INP_{-10}$ to air masses above the Alps, at least during summer and autumn. During periods with admixture of air from the planetary boundary layer (PBL), particles emitted from local and regional sources in the airshed of JFJ can be transported to the high-altitude site. In summer, such PBL injections occur frequently and are mainly driven by thermally induced processes such as convective PBL growth, anabatic mountain winds and mountain venting (Collaud Coen et al., 2011; Griffiths et al., 2014; Henne et al., 2004; Ketterer et al., 2014; Poltera et al., 2017). Additionally, frontal systems and dynamically driven winds can lift air from the PBL above the Alps (Ketterer et al., 2014; Lothon et al., 2003).

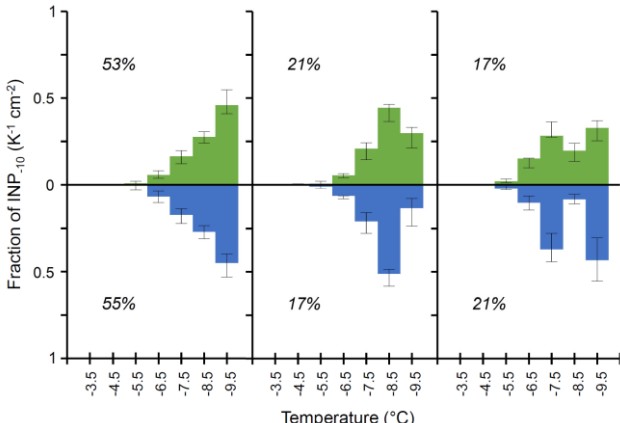

Figure 4: Median values of the three most abundant types of differential INP spectra observed on tree leaves (green) in Gempen (650 m a.s.l., n = 53) and in air (blue) at Jungfraujoch (3580 m a.s.l., n = 53), normalized to the cumulative INP-10 concentration considering the spectra with a monotonous increase in differential INP concentrations with decreasing temperature, and spectra with statistically significant peaks (section 2.3). Indicated percentages are the relative frequency of each spectral type among spectra with clearly defined patterns found on foliage and in air, respectively, error bars the 1st and 3rd quartile. Note, that the number of analysed droplets per sample is different between Gempen (144) and Jungfraujoch (52).

Our findings also suggest that the same parameter, or set of parameters, control the aggregation of IN proteins on the scale of leaves within a canopy as in the wider landscape (i.e., airshed upwind of JFJ). Such a driver could, for example, be leaf wetness duration. It varies within single canopies (Batzer et al., 2008) and within a similar range (< 4 hours to around 10 hours per day) also on the scale of entire landscapes (Asadi and Tian, 2021). Within a canopy, local differences in shading, radiative and convective cooling and wind exposure drive variations in leaf wetness duration (Batzer et al., 2008). Within the landscape, meteorological parameters such as e.g. RH and potential evaporation are good predictors of leaf wetness duration (Asadi and Tian, 2021; Gleason et al., 2008). The effect of RH on INP-10 (Fig. 1) hints at moisture not only affecting the size of epiphytic bacterial populations (Caristi et al., 1991), but also INP density in the phyllosphere. Consequently, varying availability of moisture could possibly provide part of the explanation for both, temporal trends (Fig. 1) and differences in INP populations between samples (Fig. 2) observed in this study. Indication for the influence of environmental conditions on biological INP populations aligns with observed differences in the abundance of leaf litter derived INPs between climatic zones (Schnell and Vali, 1976) as well as the influence of meteorological parameters on INA bacteria colonizing crops (Hirano and Upper, 1989; Leben, 1988). The similarity in spectral types and their relative abundance between tree canopies and air at JFJ further suggests a certain consistency in freezing spectra between vegetation types covering a large share of terrestrial surface, e.g. forest, grassland and agricultural crops which also harbour INA microorganisms (Hill et al., 2014; Lindemann et al., 1982; Lindow et al., 1978a). Thus, the similarity of INP abundance and spectra among the four investigated tree species might apply to a broader range of trees and other growth forms.

**4 Conclusion**

To conclude, our results indicate that changes in meteorological parameters such as RH and possibly temperature, by affecting the leaf microhabitat, impact the concentration and perhaps activity of INA microorganisms on plant canopies. The similarity in spectral types and their relative abundance between tree canopies upwind of JFJ and in air at JFJ are a novel type of evidence for plant surfaces being a major source of biological INPs at cloud height above the Alps. If increasing RH or decreasing temperature lead to an enhanced INP supply at plant surfaces, the flux of biological INPs from the phyllosphere to the

atmosphere might be elevated under these conditions even in the absence of additional emission mechanisms. Therefore, at locations in the atmosphere where mixed-phase clouds can form and INPs originating from the phyllosphere comprise a large part of the biological INP population, changes in meteorological conditions, beyond rainfall (Mignani et al., 2021) could impact the INP source and, thereby, cloud development. Further exploration and quantification of the effect of meteorological parameters on biological INP populations on leaves might reveal interesting insights into the dynamics of the INP distribution

at mixed-phase cloud height.

**Data availability**

Leaf data discussed herein is provided in the supplement of this article. Data collected at Jungfraujoch is available upon request from the corresponding author.

**Autor contribution**

AE and FC designed the study, AE conducted experiments at Jungfraujoch and analysed the data, AE and FC collected and analysed leaf data, AE wrote the manuscript with contributions from FC.

**Competing interests**

The authors declare that they have no conflict of interest.

**Acknowledgements**

Sampling a vertical profile of a *Fagus sylvatica* in Hölstein was possible thanks to the Swiss Canopy Crane II, an infrastructure financially supported by the Swiss Federal Office for the Environment FOEN and the University of Basel. We thank the crane operator Niek ten Cate for his support during sampling in Hölstein and Linus Keiser and Jean-Luc Mosimann for their help during the analysis of leaf carbon and nitrogen content. We are grateful to the International Foundation of the High-Altitude Research Stations Jungfraujoch and Gornergrat (HFSJG), 3012 Bern, Switzerland, for providing us the opportunity to work

and conduct experiments at the high-altitude observatory Jungfraujoch. We thank Stephan Henne for performing FLEXPART simulations for us, which we included in the reply to the general comment of Referee #2. We acknowledge financial support for this study by the Swiss National Science Foundation (SNSF), grant no. 200020-212121.

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
