# Peer review of "Similar freezing spectra of particles on plant canopies and in air at a high-altitude site"

_EGUsphere, 2024_

## Author Response (AR1)

**Author's response to RC1 and RC2**

**Response to RC1**

*We are grateful to Referee #1 for their careful and detailed comments and review. Please find below our point-by-point replies (blue font) to all comments (black font). Changes made to the manuscript are in orange font. Where modifications have been made to the original manuscript, the respective passage is highlighted in bold font. Line numbers associated with the modifications refer to the revised manuscript.*

RC1:

Einbock and Conen present a well-written manuscript on measurements of ice nucleating particles (INPs) in washing water of leaves from a number of trees and air samples collected in Switzerland. The study appears to be well-conceived and methodologically sound. The results provide valuable insights into INPs from plant surfaces and provide a solid basis for further research. It is a valuable contribution to the literature on potential sources of atmospheric INPs. Some issues still need to be addressed before publication.

Specific comments:

L15: Please clarify what is meant by "exposed leaves" with respect to the type of exposure.

We specified the sentence:

The cumulative concentration of INPs active at $\geq$ -10 °C (INP$_{-10}$) did not vary significantly between the investigated tree species, but as inferred from leaf mass per area and leaf carbon isotopic ratios seemed to be lower **on sun as compared with shade leaves**. (L13-15)

L37-38: "..and living as well as decaying vegetation (Lindemann et al., 1982; Lindow et al., 1978a; Schnell and Vali, 1976) are major sources of biological INPs." The statement needs to be modified and references added, because vegetation is defined as the plant cover in a given area. However, the references cited are for bacterial INPs. While these studies link bacteria to living or decaying vegetation, they don't directly address the vegetation itself as source of INP. Given that this study analyzed INPs from leaf washing water, the possible contribution of plants themselves as sources of biological INPs should be mentioned (e.g., Pummer et al., 2012, Hiranuma at al., 2015, Felgitsch et al., 2018., Seifried et al., 2020).

Thank you for pointing out this issue. We added the references Felgitsch et al. 2018, Haga et al. 2014, Hiranuma et al. 2015, Pummer et al. 2012 and Seifried et al. 2020 to the statement to account for the contribution of INPs associated with plant material itself to the overall biological INP population. Also, we split the references between living and decaying vegetation to make the allocation clearer:

Soil organic matter (Conen et al., 2011; Hill et al., 2016; O'Sullivan et al., 2014) and living **vegetation** (**Felgitsch et al., 2018; Hiranuma et al., 2015**; Lindemann et al., 1982; Lindow et al., 1978; **Pummer et al., 2012; Seifried et al., 2020**) as well as decaying vegetation (**Haga et al., 2014**; Schnell and Vali, 1976) are major sources of biological INPs. (L37-39)

L38-40: Please also consider the recent study by Wieland et al., 2024 that found birch INP to be active above -10°C.

Thank you for drawing our attention to this study. The revised manuscript now relates to it in several instances:

Recently, also INP$_{-10}$ originating from pollen have been identified (Gute and Abbatt, 2020; Kinney et al., 2024; Wieland et al., 2024). (L41-42)

Recently, such clustering was also discovered in cell-free INPs shed by **the** ice-nucleation active (INA) fungus *Fusarium acuminatum* (Schwidetzky et al., 2023) **and in ice nucleating macromolecules (INMs) released by pollen of *Betula pendula* (Wieland et al., 2024)**. (L54-55)

Potential additional sources of INP$_{-10}$ at the sampling locations **known at present** might include **INMs** derived from pollen (Gute and Abbatt, 2020; Kinney et al. 2024; **Wieland et al., 2024**). (L187-189)

L112: While I would expect that any counting error would be detected when two independent observers count by eye, I wonder about a potential error when only one observer counts by eye. How often has it been counted?

Frozen tubes are usually counted once at each temperature step. Only if there has been a distraction, i.e., someone starting to talk to the counting observer, the tubes are counted twice. We found that counting errors occur to new observers still in the process of developing a routine in handling and properly illuminating the samples. Yet, the learning curve is steep and after a while errors become extremely rare. That is, independent experienced observers of the same assay almost always report identical numbers of frozen tubes. We have made this experience many times with new laboratory assistants, but have not systematically recorded or analysed these tests. Observations in this study were done by the authors, who both have several years of experience. Having two observers was more to increase the speed of analysis than to detect very rare eventual counting errors. We amended the manuscript accordingly:

After every 1 °C step in cooling (rate 0.3 °C min$^{-1}$), temperature was left unchanged for at least 30 s before the number of frozen droplets was determined visually by one **observer on both subsets**, or two independent observers **on one subset each**. (L120-122)

L116: Leaves were initially collected in polyethylene zip bags and were transferred into 50 mL tubes after colour assessment. The authors should add information on the handling of the leaves for colour assessment. Were the leaves removed from the zip bags? Where were they placed? For how long? How were the leaves handled? Both, the material used and the handling of the leaves could have introduced INP contamination. Did the authors perform INP tests on the zip bag and tubes used, e.g., by washing them with MilliQ to exclude such contamination?

Thank you bringing this up. To test INP contamination of the zip bags and tubes, a blank was measured by filling 50 mL of ultrapure water (W4502-1L, Sigma-Aldrich) into a zip bag, shaking it so that the entire bag surface was wetted and filling it into a 50 mL tube as used for leaf analysis (Cellstar®, greiner bio-one, Switzerland). The subsequent INP analysis was performed as for the leaf samples. Tubes only froze several Degree Celsius below -10 °C:

Blank measurements had shown the zip bags did not contain INP$_{-10.}$ (L84-85)

We further complemented the revised manuscript with information on handling of the leaves during colour assessment:

For the colour assessment which took a few minutes per sample, leaves were removed from the polyethylene zip bags and spread adaxial side up on top of a new polyethylene zip bag cleaned with 2-Propanol. (L103-105)

Throughout the process, leaves were handled with gloves (Vasco® Nitril light, B. Braun, Switzerland) cleaned with 2-Propanol. (L107-108)

L126: Were all standards used in all calibrations? Can the authors add more details about the standards used, e.g., company, reference?

We are sorry for the imprecision and thank you for commenting on this. No, we did not use all standards in all calibrations. Specific Glycine (Arndt Schimmelmann Glycine No 4, https://hcnisotopes.earth.indiana.edu/doc/alphabetical-list-of-all-reference-materials-ada.pdf) and USGS62 (Coplen, 2019b) were used for additional analyses not presented in the final version of the manuscript.

We specified the section and added references to the used standards:

Mass calibration for C and N quantification was performed with the lab standard EDTA (41.09% C; 9.59% N). For isotopic (C) size correction and calibration the lab standard EDTA (41.09% C; 9.59% N), USGS61 (Coplen, 2019b) and USGS40 (Coplen, 2019a) were used. (L135-137)

L135: "Usually" implies that some samples were collected differently. No number is given, but does this refer to "The remaining four samples" mentioned later? Consider rephrasing for more clarity.

Yes, the "usually" refers to the majority of 129 samples collected during 25 min of impinger operation compared to four samples collected during 15 min of impinger operation.

We reformulated the two sentences for more clarity:

The majority of samples (n = 129) were collected throughout 30-min periods consisting of 5 consecutive 5-min intervals. For the remaining samples (n = 4), the impinger was operated at three consecutive 5-min intervals. (L146-148)

L138: Was this also done after the last 5-min sampling interval? If yes, I suggest to write " ..was replenished after each 5-min sampling interval"

No, water was not replenished after the last 5-min sampling interval. Instead, the volume of the remaining liquid in the sampling cone was quantified.

We expanded the description in the manuscript:

After the last 5-min interval, liquid volume in the sampling cone was quantified. (L149-150)

L162: The authors found a correlation between INP concentration and relative humidity (RH). Given that RH and rainfall are related, I wonder about the potential influence of rainfall events on this observed correlation. For example, Bigg et al. (2015) have reported a persistent effect of rainfall on INP concentrations. It would be interesting to explore this relationship for this dataset by plotting INP concentration against rainfall frequency, if such data are available.

We agree and had also expected to potentially find a correlation between rain events and INP concentrations. However, precipitation measurements at Gempen obtained from the data portal of the Swiss Federal Office of Meteorology and Climatology (MeteoSwiss) showed no correlation between total precipitation (mm) and INP concentrations for different time spans (within 24 h, 48 h, 7 days and 14 days prior to sampling, measurement intervals ending at 0600 UTC on the sampling day):

| date | median $INP_{-10}$ concentration | Sum of precipitation (mm) in preceding time interval | | | |
|---|---|---|---|---|---|
| | | 24 h | 48 h | 7 d | 14 d |
| 09.08.2023 | 3.8 | 0 | 2.7 | 16.2 | 28.0 |
| 22.08.2023 | 7.0 | 0 | 0 | 0 | 15.8 |
| 12.09.2023 | 9.5 | 0 | 0 | 0 | 8.2 |
| 26.09.2023 | 16.1 | 0 | 0 | 10.3 | 64.0 |
| 03.10.2023 | 28.6 | 0 | 0 | 0 | 10.3 |
| 17.10.2023 | 20.6 | 0 | 0 | 7.0 | 7.0 |
| 30.10.2023 | 46.8 | 0 | 0 | 48.8 | 64.6 |
| 15.11.2023 | 38.0 | 17.3 | 30.0 | 62.1 | 128.6 |

Since rainfall can impact biological INPs and also in response to a comment by Referee #2 on the same topic, we expanded the manuscript:

There was no precipitation 24 h prior to sampling during the entire campaign at both sites, except for the last sampling day in November (17.3 mm over 24 h). (L176-177)

L172-174: The statement regarding the contribution of pollen to INP concentrations should be explained. The authors do not present data to support the absence of pollen, pollen fragments, or pollen-derived INP in their samples. While the four tree species studied do not appear to pollinate during the seasons studied, other tree species, flowers, and shrubs - including some for which IN activity may not be known - may do so. Pollen could also come from long-distance transport from other locations.

We understand that the potential contribution of pollen to the INP population should be discussed further in the manuscript. We now do this based on continuous measurements of atmospheric pollen concentrations from the nearest monitoring station operated by MeteoSwiss about 10 km NW of Gempen:

Tree species commonly found in central European forests, including the four species investigated here, typically do not flower between August and November (Anonymos, 2002). Still, wind-dispersed pollen from other plant species and long-range transport deposited onto the leaves might have contributed to the analysed INP population. Atmospheric pollen concentrations at Basel are continuously monitored and measurements for about 50 plant species or families are available until 29.09.2023, seven among them until the end of the sampling period. While cumulative $INP_{-10}$ concentrations on leaves increased, airborne pollen concentrations showed a decreasing trend. There was no correlation between pollen concentrations from individual species or families and $INP_{-10}$ concentrations. In the following, we will therefore focus on microbial INPs. (L189-196)

L184/185: Consider including results on temperature exposure of different tree species in the main text or supplementary section of the manuscript.

Unfortunately, there is no data available on the leaf temperature of the sampled trees.

L185/186: Clarify what "different exposure of the trees" refers to. Exposure to what?

We reformulated the paragraph for clarification:

The significant correlation between air temperature and $INP_{-10}$ concentration in *P. avium* and the absence of such a correlation in the other species might have been related to **the free-standing position of *P. avium* in a meadow while the other trees investigated were situated within a forest. Thus, the *P. avium*** were less shielded by **surrounding trees** against radiative cooling **during clear nights** and might have experienced lower leaf temperatures, pronouncing the effect of temperature on the observed $INP_{-10}$ concentrations. Jordan and Smith (1994) found leaf temperatures during clear nights were around 4°C below air temperature. (L206-211)

L211: Clarify what the asterisk and the slash mean in this context

In this context, the asterisk and slash refer to the mathematical operations of multiplication and division. Analogous to the description of normally distributed data with arithmetic mean and additive standard deviation (arithmetic mean $\pm$ additive standard deviation), the use of the geometric mean and multiplicative standard deviation has been proposed for the description of log-normally distributed data (geometric mean $^{x}/$ multiplicative standard deviation). It means that 68% of the values in a distribution are between the geometric mean divided by the multiplicative standard deviation and the geometric mean multiplied by the geometric standard deviation (Limpert et al., 2001).

We should have used an *x* as symbol for multiplication instead of the * for more clarity and also better explain the underexploited multiplicative standard deviation:

In the vertically sampled *F. sylvatica* (HOL), $INP_{-10}$ concentrations increased from the top to the lowest part of the canopy. The four samples from the top had a median value of 4.0 $INP_{-10}$ $cm^{-2}$ with a multiplicative standard deviation of $^{x}/$ 1.5 (i.e., 68% of leaf samples at the top of the tree had an $INP_{-10}$ concentration between 2.7 (4.0 / 1.5) and 6.0 (4.0 x 1.5) $INP_{-10}$ $cm^{-2}$). (L234-236)

L215: Please clarify what "this position" refers to

We rephrased the sentence to clarify that "this position" refers to the sample collected from the bottom part of the canopy facing a SE direction:

Five weeks later, INP concentrations at **the lowest position facing a SE direction** had decreased **distinctly from 73.1 $INP_{-10}$ $cm^{-2}$ to 15.8 $INP_{-10}$ $cm^{-2}$** but were still twice as high as at the opposite canopy side. (L239-240)

L253: For completeness, it may be helpful to provide a brief explanation of what Type II INPs are. Also, the reference provided supports this classification for bacterial INPs, but not for biological INPs, which include fungal or plant INPs.

We added an explanation for Type II INPs and changed "biological" to "bacterial" INPs to match the cited reference.

According to their efficiency, **bacterial** INPs are categorised as either type I, II or III, corresponding to a decrease in nucleation temperature. (L276-277)

Type II INPs are typically less abundant and induce freezing between -5 °C and -7 °C (Yankofsky et al., 1981). (L277-278)

L258: Please add specification of what "leaf habitat properties" might involve (e.g., microclimate, leaf morphology, etc.).

Done:

These differences indicate that variations in leaf habitat properties **such as microclimate, leaf morphology and physiology, and cooccurring differences in the phyllosphere microbiome** between tree species might have contributed variation to the distribution of spectral types among species. (L282-285)

L275: It may be useful to clarify that "radiation" refers to solar radiation or UV exposure to avoid ambiguity.

We complemented the sentence accordingly:

For example, **UV and γ-**radiation (de Araujo et al., 2019; Govindarajan and Lindow, 1988), desiccation (de Araujo et al., 2019), decreasing pH (Lukas et al., 2022) and high temperatures around 30 °C (Nemecek-Marshall et al., 1993) seem to trigger the dissolution of IN protein complexes, whereas lower temperatures around 15 °C and nutrient limitation can promote their formation (Nemecek-Marshall et al., 1993; Ruggles et al., 1993). (L301-305)

L280: The statement "The more efficient an INP, the more sensitive it is to stress (Govindarajan and Lindow, 1988)." does not seem to apply universally to all types of INPs and stresses (e.g., Kunert et al., 2019, Eufemio et al., 2023).

This is correct, we have to be more precise here:

The **larger an IN protein cluster and the higher the resultant nucleation temperature of the proteinaceous INP, the faster its activation temperature is lowered by certain stressors, e.g. radiation** (Govindarajan and Lindow, 1988). (L306-307)

L283/284: This is rather speculative. Fagus sylvatica could simply harbor different INPs, such as those associated with specific plant pathogens. Certain plant pathogens can contribute to the diversity of biological INPs (e.g., Morris et al., 2008, 2013, Kunert et al., 2019).

Plant pathogens are typically host-specific, meaning that they are adapted to infect particular plant species. While they can occasionally be found on non-host plants due to factors like accidental contamination or environmental conditions, they do not cause disease or reproduce effectively on these plants. This host specificity suggests that different plant species might host distinct INP-producing microorganisms or pathogens that are not present or are less prevalent on other species.

Thank you for the instructive comment. We agree that the described mechanism is only one among several conceivable explanations for the development of the monotonous spectral type which we emphasize in the revised manuscript:

The gradual breakup of larger IN protein clusters under continued stress in unfavorable conditions, or the inhibition of INP cluster formation could be **one** possible explanation for the development of the monotonous spectral type. This **explanation would imply** that conditions for the expression and aggregation of IN proteins were less suitable on *F. sylvatica* as compared to the other investigated tree species. **Another explanation could be host-specific properties of *F. sylvatica* resulting in a limited set of INP-producing microorganisms that generate the monotonous spectral type only.** (L308-313)

L286/287: Please provide an explanation or context for how these factors influence INP behaviour?

The availability of water and nutrients critically influences the survival and growth of microbial populations in the phyllosphere (Vorholt, 2012). The mentioned factors (cuticle characteristics, leaf wettability, leaf exudates and leaf topography) can impact the availability of water or nutrients on leaf surfaces as well as differ between tree species. Thus, the mentioned factors could potentially influence the population size or composition of epiphytic (INA) microorganisms or, if the availability of nutrients is being affected, their nucleation frequency (Nemecek-Marshall et al., 1993; Ruggles et al., 1993).

We added the following explanation in this context:

Yet, other differences in microhabitat, for example due to variations in cuticle characteristics, leaf wettability, leaf exudates or leaf topography (Yan et al., 2022), might have contributed to the unequal distribution of spectral patterns between tree species. **These factors can impact the availability of water or nutrients on leaf surfaces, thereby affecting population size and perhaps also composition of epiphytic (INA) microorganisms. In addition, there is a link between nutrient availability and the frequency at which ice-nucleating activity is expressed in certain INA-species (Nemecek-Marshall et al., 1993; Ruggles et al., 1993**). (L314-319)

Figure 3: Clarify what is meant by "dark backgrounds" when all backgrounds are in light-colors.

In Figure 3, we distinguish seven different panel background colours: white as well as a lighter and a darker version each for orange, blue and grey. However, since all colours are rather light and visual differentiation between them could be challenging, we decided to revise Figure 3 by assigning more vibrant colours as panel backgrounds.

Also, we specified the caption of Figure 3:

Colours reflect leaf colours, panel background **colour** indicates the spectral type (white: monotonous spectral type, **brown**: peak at -8.5 °C, **green**: peak at -7.5 °C, **purple**: peak > -7.5 °C). **Dark brown, dark green** and **dark purple** panel backgrounds are for spectra with significant peaks, **light brown, light green** and **light purple** panel backgrounds are for spectra with insignificant peaks (section 2.3). (L291-293)

L334-336: Please specify what dynamics refers to (e.g., distribution, activity, concentration).

We modified the last two sentences of our conclusion and specified what dynamics refer to:

Therefore, at locations in the atmosphere where mixed-phase clouds can from and INPs originating from the phyllosphere comprise a large part of the biological INP population, changes in meteorological conditions **beyond** rainfall (Mignani et al., 2021) **could** impact **the INP source and, thereby,** cloud development. Further exploration and quantification of the effect of meteorological parameters on biological INP populations **on leaves** might reveal interesting insights into the dynamics **of the INP distribution** at mixed-phase cloud height. (L368-373)

Technical corrections/typos:

L36: remove the period after "thereof"

Done.

L76: Form-> From

Corrected.

L85: remove space before "100"

Done.

L91: NSC->NCS

Corrected.

L113/115: I assume the authors mean "dilutions" instead of "dilution series" in both instances

Yes, thank you for spotting this.

L114: were -> where

Corrected.

L118: concentration -> concentrations

Corrected.

L123: was -> were

Corrected.

L127: analyszed -> analysed

Corrected.

L131: Consider rephrasing to " JFJ is at a 3 km higher elevation than the foilage sampling sites"

We rephrased the sentence:

The observatory JFJ is located at a 3 km higher elevation **than the foliage sampling sites and situated** about 110 km SSE **of Gempen**. (L142-143)

L159/160: I think it should read "per cm$^2$" of leaf area

Changed accordingly.

L236: Consider rephrasing the last part of the sentence to " leaves that are more exposed to sunlight in the canopy"

We rephrased the statement and now mention that leaves exposed to more sunlight in a canopy might be more exposed to a number of different stressors potentially removing INPs more efficiently than from leaves in other parts of the canopy in an additional sentence:

Our results from Hölstein indicate that INP$_{-10}$ concentrations are lower on **leaves more exposed to sunlight in a canopy. Such leaves at the upper and outer part of a canopy might be in general more exposed to stressors or processes potentially removing INPs from leaves compared to leaves in other parts of the canopy.** (L257-260)

L247: „warmer" instead of „colder" temperatures; the values given afterwards are higher than the values given in the sentence before

Thank you for noticing this.

L332: from -> form

Corrected.

Table S1/S2 captions: Add a definition for LMA

Done.

Table S2 caption: leaf -> Leaf

Corrected.


[revised manuscript text omitted]

Author's response to RC2

*We are grateful to Referee #2 for their thoughtful and valuable review and comments. Please find below our point-by-point replies (blue font) to all comments (black font). Changes made to the manuscript are in orange font. Where modifications have been made to the original manuscript, the respective passage is highlighted in **bold font**. Line numbers associated with the modifications refer to the revised manuscript.*

**Review:** Similar freezing spectra of particles on plant canopies as in air at high-altitude site

**Summary:**

In this manuscript, Einbock and Conen investigate tree surfaces as sources of atmospheric ice-nucleating particles (INPs) active above -10C. Specifically, they analyzed the freezing spectra of washing water from leaves of four different tree species that are abundant in Switzerland. The authors identified recurring freezing patterns and compare these with air INP samples collected at a high-altitude mountain station in the Alps (Jungfraujoch). Their findings reveal a correlation between the freezing patterns of leaf washing water and the air samples, suggesting a potential link between plant-derived INPs and those found at high altitudes.

This study is of interest to the INP community, offering valuable insights into the potential sources of biological INPs. The connection between the plant surface INPs and INPs collected at high altitude is particularly intriguing, contributing to ongoing discussions about the origins of biological INPs. The manuscript is well-structured and merits publication in Biogeosciences after addressing some minor revisions.

**GENERAL COMMENTS:**

While the manuscript offers a compelling analysis, a discussion on the transport mechanisms of INPs from leaf surfaces to high altitudes (Jungfraujoch station) is notably absent. Although this topic may not be the primary focus of the manuscript, it is crucial to address how these INPs could be transported to such altitudes, especially if the claim is that they originated from plants at lower elevations.

Furthermore, did the authors consider analysing the back trajectory of air masses during the field campaign? This would help clarify whether the INPs sampled at the station could indeed have originated from local vegetation, or if they were transported from distant sources.

Aerosol particles emitted from local and regional sources can reach the Jungfraujoch (JFJ) periodically with air from the planetary boundary layer (PBL). Periods influenced by PBL injections and free tropospheric conditions can be distinguished at JFJ by continuous radon-222 measurements (Griffiths et al., 2014; Herrmann et al., 2015). Based on a threshold radon-222 concentration of 0.64 Bq m$^{-3}$ (Conen and Zimmermann, 2020), about 86% of our JFJ samples were collected during periods with admixture of air from the PBL (radon data obtained from https://sievju.org/pl/radon.pl). During summer, PBL injections at JFJ are mainly driven by thermodynamic processes such as anabatic mountain wind, convective PBL growth and mountain venting, especially during anticyclonic conditions (Collaud Coen et al., 2011; Henne et al., 2004; Ketterer et al., 2014; Poltera et al., 2017). Additionally, frontal systems and dynamically driven winds can transport air from the PBL above the Alps (Ketterer et al., 2014; Lothon et al., 2003).

To investigate the potential source regions of the collected INPs, Stephan Henne (Swiss Federal Laboratories for Materials Science and Technology (Empa)) has simulated for us source sensitivities

for the sampling days using the Lagrangian dispersion model FLEXPART, configuration COSMO (Pisso et al., 2019). Simulations were initiated above the JFJ at 3400 m a.s.l. Starting at 00:00 UTC, 50'000 inert particles were released every three hours and traced back for 10 days. Please find below eight examples of the visualised simulation results (three hour means, time displays end of three-hour simulation intervals). Results indicate that near-ground (below 50 m a.g.l.) residence time of the simulated particles was longest in Switzerland and its neighbouring countries, suggesting that terrestrial surfaces in this area were important potential sources of the sampled INPs during the campaign.

[Figure]

We added a discussion on how the sampled INPs could have been transported to the JFJ to the revised manuscript. Since transport mechanisms are not the main focus of our manuscript, we prefer to present the FLEXPART output only in our reply to your comment:

During periods with admixture of air from the planetary boundary layer (PBL), particles emitted from local and regional sources in the airshed of JFJ can be transported to the high-altitude site. In summer, such PBL injections occur frequently and are mainly driven by thermally induced processes such as convective PBL growth, anabatic mountain winds and mountain venting (Collaud Coen et al., 2011; Griffiths et al., 2014; Henne et al., 2004; Ketterer et al., 2014; Poltera et al., 2017). Additionally, frontal systems and dynamically driven winds can lift air from the PBL above the Alps (Ketterer et al., 2014; Lothon et al., 2003). (L331-336)

**SPECIFIC COMMENTS:**

Title:

- Consider changing 'as' to 'and' for better grammar: 'Similar Freezing Spectra of Particles on Plant Canopies and in Air at a High-Altitude Site.'

Thank you for the suggestion. We changed the title accordingly.

Abstract:

- Line 8: The phrase "efficiently freeze" is vague. Consider rephrasing to specify whether you mean freezing at higher temperatures or some other criterion.

Thank you for pointing this out. We deleted the word "efficiently" as it refers to higher temperatures ("slightly supercooled cloud droplets") mentioned later in the sentence:

Certain aerosolized microorganisms are able to freeze slightly supercooled cloud droplets and therefore affect mixed-phase cloud development. (L7-8)

Introduction:

- The introduction is well-written and provides a comprehensive background on the topic.

Thank you.

- Line 77: typo 'form' should be 'from'

Corrected.

Methods:

- Line 106: Sonicating leaves may release INPs from within the leaf structure, not just the surface. This could lead to an overestimation of surface-derived INPs. Did the authors compare results with and without sonication to assess its impact on the data?

Thank you for commenting on this. Prior to the study, we had compared results after different durations of sonication, but not of sonication and no sonication. Our starting point were the experiments of Buesing and Gessner (2002). They had found sonication to be severalfold more efficient in detaching bacterial cells from leaf litter than vortexing. Sonicating for 1 min was a compromise between getting the majority of epiphytic microorganisms into suspension and avoiding visible leaf damage potentially releasing INPs from within the leaf structure. Pre-trials with *Syringa vulgaris* leaves had shown us that only around one-third of $INP_{-10}$ was removed after 10 s of sonication, whereas after 6 min leaf surfaces had suffered visible damage. Three-quarters of all $INP_{-10}$ found after 6 min sonication had already been removed after 1 min of sonication, when the leaf surface had not yet suffered visible damage. In other words, sonication between minute 1 and 6 opened the leaf surface with many small bruises but brought three times less INPs into suspension than did the first minute of sonication. Whether the additional INPs released between minute 1 and 6 were from the leaf surface or from within the leaf structure could not be assessed.

In response to your question, we returned to the issue and analysed a fresh *Tilia platyphyllos* leaf collected on 26. Aug. 2024 from nearby our laboratory, analysed it a first time as we had analysed all leaves mentioned in the manuscript. We then rinsed the leaf with deionised water and analysed it a second time by the same procedure. After the second sonication we found only one-seventh of the amount of $INP_{-10}$ released after the first sonication. No damage was observed on the leaf after the second minute of sonication. In case these additional INPs had come from within the leaf structure, some within-leaf-structure-INPs may already have been released during the first minute of sonication and our measurements would be an overestimate of leaf-surface-INPs. If the INPs released during the second minute of sonication originated from the leaf surface, we would have underestimated the INP concentration on the leaf surface by sonicating for 1 min only. Although we are not in a position to judge which is the right interpretation or whether both apply to some extent, we can say that either way the effect on the overall results would be minor.

We complemented the revised manuscript with a statement addressing this issue:

Sonication for 1 min was a compromise informed by additional trials. When compared to different sonication times between 10 s and 6 min, 1 min was long enough to suspend the majority of INPs from the leaf surface while avoiding visible leaf damage that may promote the release of INPs from within the leaf structure. (L114-116)

- Line 127: typo 'analyszed' should be 'analyzed'

Corrected.

- Line 144- 146: The explanation of INP corrections is somewhat vague. It would be helpful to expand on how the sodium concentration correlates with INP concentration, possibly with an additional equation for clarity.

We expanded on this issue:

Sodium analysis of impinger liquid (940 Professional IC Vario, Metrohm, Switzerland) revealed that its concentration decreased to 70% of its initial concentration during the 25 min the impinger was operated. **This decrease indicates some loss of liquid in form of droplets; a loss of liquid in form of water vapour only would not have removed NaCl.** Therefore, we corrected INP concentrations for potential INP loss with droplets exiting the impinger during collection by multiplying uncorrected INP concentrations with a factor of 2/(1+0.7)**, assuming the loss of INPs is zero at the beginning of sampling, when there are no INPs in the liquid, and increases linearly with time. I.e., the fraction of INPs lost with droplets is half the fraction of NaCl lost.** (L154-160)

Results & Discussion:

- Line 211: Was there any rainfall during the field campaign? If so, could rain have washed INPs from the top of the canopy to the bottom, leading to their accumulation in the lower parts of the trees, similar to the findings of Seifried et al. 2020?

We agree that there are several factors and processes that could cause INP$_{-10}$ concentrations to increase from the top of the canopy to the bottom and that rainfall might be one of them. During the field campaign, however, there was no rainfall in Hölstein within at least four days prior to sampling the vertical canopy profiles. As population sizes of epiphytic microorganisms can change distinctly on time scales of several hours (Hirano and Upper, 1989; Lindow and Brandl, 2003), we assume that in this study the influence of rainfall on the distribution of INP$_{-10}$ between the canopy top and bottom was probably minor.

Since precipitation can impact the dynamics of biological INPs and also in response to a comment by Referee #1 on the same issue, we expanded the manuscript:

There was no precipitation 24 h prior to sampling during the entire campaign at both sites, except for the last sampling day in November (17.3 mm over 24 h). (L176-177)

- Figure 1: To improve clarity, consider labeling the individual graphs as (a), (b), (c), and (d). This would allow for a more structured caption and make it easier to refer to specific graphs in the text. Additionally, using "hollow" and "filled" symbols instead of "dark" and "bright" could enhance differentiation. Assign the corresponding Spearman correlation analysis to each labeled panel.

Thank you for the suggestions. We revised Figure 1 and labelled the individual graphs as a) to d), assigned the corresponding Spearman correlation analysis to the panels and modified the caption accordingly:

Cumulative INP$_{-10}$ concentration versus leaf mass per area (LMA) (**a, c**) and leaf $\delta^{13}$C values (**b, d**) for the GEP data set (**c, d**) and all *F. sylvatica* samples (**a, b**). Colours **and symbols** represent tree species (brown **circles**: *F. sylvatica*, blue **triangles**: *P. avium*, green **diamonds**: *J. regia*, red **squares**: *T.*

*platyphyllos*), light colours for first and dark colours for second tree sampled **(c, d)**. *F. sylvatica* sampled at Hölstein is represented by squares in the top panels **(a, b)**. (L253-256)

We would like to keep the filled symbols to illustrate differences between the four tree species (the four main colours).

- Figure 3 and 4: The connection between the last pattern in Figure 4 and those in Figure 3 is unclear. Is it the blue pattern? Could you in general provide more explanation on how the patterns were selected?

[Colours mentioned in this reply refer to the preprint version of Figure 3] Yes, the last pattern in Figure 4 corresponds to the (dark) blue pattern (statistically significant peak at -7.5 °C) in Figure 3.

In Figure 3, we present differential INP spectra for the entire Gempen data set. Spectra of all samples collected are shown individually in separate panels. For the analysis in Figure 4, we only considered spectra with clearly discriminable patterns, that is spectra with a monotonous increase in differential INP concentrations with every temperature step in cooling and spectra with statistically significant peaks (section 2.3). For the Gempen samples, this corresponds to spectra with white (n = 28) and spectra with dark grey, dark blue and dark orange panel backgrounds (n = 25) in Figure 3. Samples with statistically insignificant peaks (section 2.3.) represented with light coloured panel backgrounds in Figure 3 (n = 11) were not included into the analysis on which Figure 4 is based on.

Among all significant peaks in the Gempen data set, most occurred at -8.5 °C and -7.5 °C. Significant peaks at temperatures > -7.5 °C were less frequent. For the comparison of spectral patterns between samples collected from Gempen (green) and samples collected from JFJ (blue) in Figure 4, we focus on the three most abundant spectral types in both data sets, considering, as mentioned earlier, only spectra with clearly discriminable patterns. Therefore, all samples from Gempen expressing the monotonous spectral type (white panel background in Figure 3) are included in the upper left panel of Figure 4 (green columns), all spectra with a significant peak at -7.5 °C (dark blue panel background in Figure 3) in the upper right panel and all spectra with a significant peak at -8.5 °C in the upper middle panel.

Please note that out of the 25 samples with statistically significant peaks, 23 samples exhibited one, and two samples exhibited two significant peaks: *J. regia* 1 and *T. platyphyllos* 1 collected on 15. Nov. showed a significant peak at -8.5 °C and at -4.5 °C or -3.5 °C, respectively:

The remaining 25 samples exhibited one **(n = 23)** or two **(n = 2)** significant peaks (section 2.3) at temperatures ranging from -3.5 °C to -8.5 °C (numbers indicate the centre of 1 °C temperature intervals implemented in the assays). (L267-269)

Both spectra are included in the analysis for Figure 4 due to the statistically significant peak at -8.5 °C. In Figure 3, however, their panel backgrounds are coloured according to the temperature of the warmest significant peak, an issue we have to indicate in the legend:

The asterisk marks spectra with two significant peaks: one peak >-7.5 °C and one peak at -8.5 °C. (L293-294)

Samples and therefore patterns from the JFJ data set were selected following the same procedure as for the Gempen data set and are displayed in blue columns in Figure 4.

We hope this explanation helped to clarify the connection between Figure 3 and Figure 4 and how the patterns in Figure 4 were selected. We expanded the explanation in the revised manuscript as follows:

Median **values** of the three most abundant types of differential INP spectra observed on tree leaves (green) in Gempen (650 m a.s.l., n = 53) and in air (blue) at Jungfraujoch (3580 m a.s.l., n = 53), normalized to the cumulative INP$_{-10}$ concentration**, considering the spectra with a monotonous increase in differential INP concentrations with decreasing temperature, and spectra with statistically significant peaks (section 2.3)**. (L338-341)

Additionally, to better visually separate the JFJ and GEP samples in our figures, we revised Figure 3 and replaced blue as a panel background colour.

- Line 297: What does "16INP_-10" refer to? Please clarify.

Sixteen INP$_{-10}$ refers to the median concentration of INP$_{-10}$ per cm$^2$ of leaf area that we found on foliage collected from Gempen throughout the entire campaign. Considering all samples collected at JFJ, we found the same concentration (16 INP$_{-10}$) in an equivalent of 3.2 m$^3$ of air. We rephrased the sentence for more clarity:

The median concentration of **16** INP$_{-10}$ observed per cm$^2$ of leaf surface in GEP was equivalent to **the median INP$_{-10}$ concentration** in 3.2 m$^3$ of air at Jungfraujoch. (L326-327)

Conclusion:

- Line 334: typo 'form' should be 'from'

Corrected.